

# Are anthropometric characteristics powerful markers to predict the Cooper Run Test? Actual Caucasian data

Gianluca Azzali[1], Massimo Bellato[2], Matteo Giuriato[3],
Vittoria Carnevale Pellino[3,4], Matteo Vandoni[3], Gabriele Ceccarelli[1] and
Nicola Lovecchio[5]

[1] Human Anatomy Unit, Department of Public Health, Experimental and Forensic Medicine, University of Pavia, Pavia, Italy
[2] Department of Information Engineering, University of Padua, Padua, Italy
[3] Laboratory of Adapted Motor Activity (LAMA), Department of Public Health, Experimental Medicine and Forensic Science, University of Pavia, Pavia, Italy
[4] Department of Industrial Engineering, University of Roma "Tor Vergata", Rome, Italy
[5] Department of Human and Social Science, University of Bergamo, Bergamo, Italy

Corresponding author
Matteo Vandoni,
matteo.vandoni@unipv.it

## ABSTRACT

**Background:** Cardiorespiratory fitness (CRF) is a powerful marker of cardiovascular health, especially in youth. Several field tests can provide accurate measurement of CRF, the Cooper Run Test (CRT) is generally preferred by physical education (PE) teachers and trainers. The CRT performance in adolescents has been compared to reference distance values, gender and age but the differences among the anthropometric characteristics of youth has not been evaluated. For these reasons, the aim of this study was to develop reference standards for CRT and evaluate possible correlations between biometric measurements and athletic performance.

**Methods:** This cross-sectional study involved a total of 9,477 children (4,615 girls) aged 11–14 years, freely recruited from North Italian middle schools. Mass, height and CRT performances were assessed in the morning during PE classes as scheduled (mornings-Monday to Friday). The anthropometric measures were collected at least 20 min before the CRT run test.

**Results:** We found a better CRT result in boys ($p < 0.001$), however a smaller SD in girls suggested a more homogeneous aerobic performance for girls (*i.e.*, 371.12 m *vs* 282.00 m). In addition, the Shapiro-Wilk test showed a low *p*-value ($p < 0.001$) but the effect size (0.031 for boys and 0.022 for girls) was small enough that the correction on this parameter allows a practical assumption of normality for the distributions. A visual homoskedastic distribution in both sexes is evident for both body mass index (BMI), mass and $VO_2$ peak with respect to CRT results. In addition, there were low linear correlation coefficients for both BMI, mass and $VO_2$ peak compared to the CRT results, with a R2 < 0.5 for every covariate. The only visual heteroskedastic distribution was observed in regression between distance in CRT and age at peak high velocity.

**Conclusions:** Our findings suggested that anthropometric characteristics are not powerful markers to predict Cooper Run Test results in a well-mixed, unpolarized and unbiased pool of middle school boys and girls. PE teachers and trainers should prefer endurance tests over the use of indirect formulas to predict performance.

## INTRODUCTION

Cardiorespiratory fitness (CRF) is the overall capacity of the cardiovascular and respiratory systems that provide a measure of the body's ability to deliver and use oxygen as energy source to support muscle activity during physical activity (PA) and exercise (*Tomkinson & Olds, 2008*). CRF is considered a powerful marker of cardiovascular health especially in youth to avoid long-term poor medical outcomes and to enhance disease prevention (*Ortega et al., 2008*). Regarding this, the accuracy of CRF measurement is considered relevant for researchers and sport specialists to assess the level of physical fitness and to predict health status. Usually, CRF is assessed through the measurement of or estimation of maximum oxygen uptake (VO$_2$ max) during a maximal graded exercise test. Even if the accuracy of this procedure is well recognized, this method is restricted to well-equipped laboratories due to the expensive, complicated and difficult experimental protocols (*Fox, 1973*).

On the other hand, several studies demonstrated the validity and reliability of specific field tests to determine CRF that provide a single measure and estimate the physiological responses required to perform prolonged exercise (*Ruiz et al., 2009*; *Ortega et al., 2015*). In fact, field-based fitness tests are easy to administer, involve minimal equipment, low cost and a larger number of participants can be evaluated in a relatively short period of time (*España-Romero et al., 2010*).

Nevertheless, there are several indirect field tests for the prediction and the evaluation of CRF (*Castro-Piñero et al., 2010*), Cooper's 12-min Run Test (CRT) is considered appropriate from childhood to maturity (*Ainsworth et al., 2000*; *Ayán et al., 2015*; *Penry, Wilcox & Yun, 2011*; *Giuriato et al., 2020*) and for this reason is widely used. In particular, CRT discovers similar results to Multistage Run Tests ($p > 0.05$), as reported by *Penry, Wilcox & Yun (2011)* and a value of bias correction factor (Cb) very high about accuracy of distance between test and retest (Cb = 0.994) and an effect size coefficient (ES) low as 0.059, which indicates the good repeatability of the test (*Alvero-Cruz, Giráldez Garcia & Carnero, 2016*).

Previously, some authors reported that 12-min run distances ranging from approximately 1,800 m to 2,500 m in adolescents (11–14 years old) and decreased in young that are overweight and obese (distance not exceed 1,500 m; *Penry, Wilcox & Yun, 2011*; *Weisgerber et al., 2009*) while *Lovecchio et al. (2013)* found an opposite trend between boys and girls. Boys showed constant performance from 11 to 13 years old (CRT distance average 2,026 m) where height and weight slightly increased with a peak at 14 years old where height and weight increased 5% and 11% respectively. In contrast, girls showed a higher performance from 11 to 12 years old (CRT distance average 1,850 m) where height and weight slightly increased and a lower performance from 13 to 14 years old (CRT distance average 1,700 m) where height and mass increased 5% and 11% respectively. More in depth, *Giuriato et al. (2020)*, through an allometric analysis of the same test within the same aged samples suggested that height positively influenced the CRF performance.

Also *Kemper (1985)* did not find improvements in CRT among 12 to 14-year-old girls (*Kemper, 1985*) and this phenomenon can be explained by the delay in growth of about two years between girls and boys (Peak of Height Velocity in Girls 11/13 years *vs* Peak of Height Velocity in Boys 13/15 years) (*Malina et al., 2021*; *Giuriato et al., 2021*). Furthermore, a study conducted by *Kirchengast (2010)* suggests that boys generally have a higher amount of lean body mass compared to girls at prepuberal age ($p < 0.001$), which allows boys to reach better cardiovascular fitness levels than girls (*James et al., 2019*).

Usually, field-based evaluations are performed by trainers or PE teachers with a common opinion that taller children are faster and lower children are stronger (*Lovecchio & Zago, 2019*). However, anthropometric measures have often been used to normalize performance (*Rowland, Vanderburgh & Cunningham, 1997*; *Welsman & Armstrong, 2000*) since recent authoritative reports have disclaimed individual normalizations based on weight or height (*Armstrong & Welsman, 2019a*) suggesting that performance also depends on growth.

Indeed to compare and evaluate CRT performance, we have to consider the divergence among ethnicity and mostly the anthropometrics characteristics such as height and mass that are specific human growth factors that affect the performance (*Lovecchio & Zago, 2019*; *Lloyd et al., 2014*): unfortunately, to the best of our knowledge, in Italy there are no findings of CRT among healthy adolescents considering the anthropometric characteristics that could be relevant also for other Caucasian countries (*Lovecchio et al., 2015*, *2019*, *2020*).

As several studies report, considering activities in the gym or on the track, anthropometric characteristics are strongly recommended to guide training, predict performance, and identify talents (*Lombardi & Piacentini, 2019*; *Buchheit & Mendez-Villanueva, 2014*; *Norton & Olds, 1996*). For these reasons, the aim of this study was to evaluate possible correlations between biometric measurements and athletic performance in middle school students and the magnitude of the influence of anthropometric characteristics on performance.

## MATERIALS AND METHODS

### Subjects

This cross-sectional study involved children from middle school and first year of middle school (11–14 years) of both sexes, sample size has been calculated using Raosoft's Sample Size calculator online tool (*Raosoft, 2004*) setting 5% as margin of error, 95% as confidence level, 2,279,335 as population size, that is the total amount of boys and girls aged between 11 and 14 in 2022 as reported by ISTAT (*Istat, 2022*), and 50% response distribution (as suggested by the tool); to be more conservative, the sample size was also calculated manually as S = $\dfrac{z^2 p(1-p)}{e^2} \Big/ 1 + \dfrac{z^2 p(1-p)}{e^2 N}$ with $N$ population size, $z$ z-score corresponding to 99% confidence interval (*i.e.*, 2.576), $e$ margin of error of 2% and $p$ experimental standard deviation of the CRT results equal to 16%, resulting in a required amount of 358 and 2,228 required subjects, respectively. However, we believe that a larger

amount of data is necessary, or at least helpful, to minimize possible bias or noise involved in the analysis.

The inclusion criteria were the possession of a valid medical certificate, to be considered as sedentary, as not implicated in activities that do not increase energy expenditure above the resting level (*European Commission, 2014*) as resulted from the school registration form. The exclusion criteria were to be involved in competitive level sports outside school and to be affected by neurological, orthopedic or cardiovascular diseases, which did not allow to run.

Overall, a final number of 9,477 participants (4,615 girls) were enrolled in the study.

Each feature of the experimental design was approved by the institutional review board of Regione Lombardia (D.g.r. 9 giugno 2017—n. X/6697) along with the Italian National Olympic Committee (CONI) and conducted in accordance with the World Medical Association Declaration of Helsinki (*JAVA, 2013*), as revised in 2018. Written informed consent was obtained from the parents or legal guardians, while verbal assent was obtained from the children after having explained to them the general purpose of the study. All participants were free to withdraw their participation at any time while it was specified that no extra academic credits were awarded for their inclusion in the samples.

## Procedures

The data collection consisted of an endurance assessment, Cooper Run Test; (*Cooper, 1968*) and anthropometric measurements.

All the procedures were conducted by a team of eight students of the sport science degree course during curricular PE classes scheduled between Monday-to-Friday in the morning (8:00–12:00 a.m.) that allow guaranteed the previous night as recovery and at least one and half hour after the last meal.

Previous training of the operators was performed to ensure the accuracy and repeatability of the procedure (inter- and intra-examiner ICC of 0.96 and 0.98, respectively) while the presence and collaboration of the curricular PE teachers were guaranteed at any time to meet the confidence and the compliance of the students (*Ceccarelli et al., 2020*). In particular, all students performed a warm-up according to their personal habits.

## Anthropometric measurements

Measurements of height and mass were taken according to the standard procedures described by the International Society for the Advancement of Kinanthropometry (*Marfell-Jones et al., 2006*). Height was measured with a stadiometer (Seca 213; Seca GmbH & Co., Hamburg, Germany) to the nearest 0.1 centimeter (cm) with participants barefooted standing in upright position with the head in the Frankfort plane. Mass was measured to the nearest 0.1 kilogram (kg) with an electronic scale (Seca 864; Seca GmbH & Co., Hamburg, Germany) with the subject wearing minimal clothing. All measurements were collected at the beginning of the PE classes (120 min of duration) ensuring at least 10 min before the successive warm-up. The anthropometric measures were not collected after the run test.

**Table 1 Equations and references for the choosen outcomes.**

| Feature | Equation | Reference |
|---|---|---|
| BMI | Quetelet Index (Mass (kg)/Height (m)$^2$) | *Ceccarelli et al. (2020)*, *Cossio-Bolaños et al. (2018)*, *Khosla & Lowe (1967)* |
| Mean velocity (km/h) | Mean Velocity = Distance covered (m)/Time (720 s) × 3.6 | |
| VO$_2$ max | Distance (km) × 22.35 − 11.29 | *Cooper (1968)* |
| VO$_2$ peak | (VO$_2$ peak (female) = [22.5 × Height (cm) − 1837.8]/Mass (kg) VO$_2$ peak (male) = [43.6 × Height (cm) − 4547.1]/Mass (kg) | *Cooper et al. (1984)* |
| Mat Offset | Mat Offset (female) = −7.7 + [0.004 × Age × Height (cm)] Mat Offset (male) = −7.9 + [0.004 × Age × Height (cm)] | *Moore (2015)* |
| Age at PHV | Age at PHV = Age − Mat Offset | *Moore (2015)* |
| PHV | The rounding of Maturity Offset | *Moore (2015)* |

Note:
BMI, body mass index; VO$_2$ max, relative maximum oxygen uptake capacity; VO$_2$ peak, peak of oxygen consumption VO$_2$ peak; Mat Offset, maturity offset; Age at PHV, age at peak of height velocity; PHV, peak of height velocity.

## Endurance cooper test

The CRT (12 min) is usually used as a preliminary and simple method to assess aerobic endurance (*Penry, Wilcox & Yun, 2011*; *Weisgerber et al., 2009*; *Cooper, 1968*).

The physical test was selected since it is strictly defined, free from operator influence, simple to administrate, cheap and easily to be organized in the school setting (*Giuriato et al., 2020*; *Carnevale Pellino et al., 2020*; *Valarani et al., 2020*).

A path along the garden of the school (minimum of 350 m) was marked every 10 m. The time was recorded from the start of running, and the distance traveled was recorded at the end of the 12 min period, which is the primary outcome of the test. All subjects were requested to run at their best to cover the maximum distance avoiding frequent acceleration/deceleration or walking phase (even if permitted) while a mate (one-to-one) checked the path and the progression of the meters run. A familiarization about the procedure was conducted 1 week before considering that the Cooper test is largely used within Italian school curricula and well known by students. All data were collected anonymously.

## Statistical analysis

Six primary features were collected: ID (text—Alphanumeric code identifying the subject); sex (categorical—M/F); age (Numerical—11/12/13/14 years); height (Numerical—height in centimeters); mass (Numerical—mass in Kilograms with one significant digit) and the target CRT result Distance (numerical—distance run in meters). In addition to primary ones, seven derived features were further added as detailed in Table 1.

The Orange toolbox (*Demšar et al., 2013*) has been adopted for the feature ranking *via* Univariate Regression or R-Relief-F (to include categorical features), to evaluate the existence of possible relationships and dependencies between the result of the CRT (*i.e.*, the distance attribute) and the other covariates.

The shapiro-Wilk test (*Kibria et al., 2022*) and Lilliefors test were used to analyze the normality of the sample, in addition statistical power analysis has been evaluated through

**Table 2 Sample numerosity.**

| Age (years) | Boys | Girls | Total |
|---|---|---|---|
| 11 | 1,188 | 1,127 | 2,315 |
| 12 | 1,435 | 1,435 | 2,811 |
| 13 | 1,394 | 1,345 | 2,739 |
| 14 | 904 | 708 | 1,612 |

the calculation of effect size (ES). Statistical analyses were performed using the Orange data mining toolbox, version 3.32 (*Demšar et al., 2013*).

# RESULTS

The average age of the participants was 12.38 ± 1.03 years and most participants were boys (52%); the age most represented was 12 years (2,811 subjects) and the least was 14 years (1,612 subjects) as represented in Table 2 (*Yıldırım Usta et al., 2019*).

Distribution of the attributes were analyzed, keeping track of the sex covariate by representing data with different colors (Fig. 1); while the Shapiro-Wilk test was run to further validate normality, providing very small $p$-values ($p < 0.001$ both for boys and girls), the effect size (0.031 for boys and 0.022 for girls) was small enough that the correction on this parameter allows a practical assumption of normality for the distributions. Analogous results were achieved *via* Lilliefors test, and graphical results are attached despite of the large and comparable size of the cohort for both boys and girls (Fig. 2), CRT results can be well fitted with Gaussian distributions, with boys showing an average higher level of performance than girls (*i.e.*, 2,096.36 m *vs* 1,807.37 m); on the other hand, a smaller SD (*i.e.*, 371.12 m *vs* 282.00 m) can be observed for girls, as represented in Fig. 3.

Parameters were ranked for information provided through univariate regression (Table 3, girls first column, boys second column) or R-Relief-F (third column) when including sex as covariate. Notably, as shown in Table 3, $VO_2$ for girls is more informative than BMI, underlying a possible causality related to different level of metabolic/muscular performance rather than structural feature like BMI. Linear regressions were performed using the four more representative covariates: BMI (Fig. 4); $VO_2$ Peak (Fig. 5); mass (Fig. 6) and age at peak height velocity (Fig. 7) despite the first is already dependent on mass and height, while the last on height. As shown in Figs. 4–7, all the R2 values were below 0.5, highlighting the absence of correlation or any sort of dependence.

Homoskedasticity was verified graphically, by plotting normalized residuals against CRT results (Fig. 8). Finally, the independence of the observations was implicitly verified by the experimental design.

# DISCUSSION

The purpose of this study was to evaluate, in a sample of middle school children, the correlation between cardiorespiratory fitness and anthropometrics characteristics, in accordance with biometric measurements and performance, as a predictor of mean results.

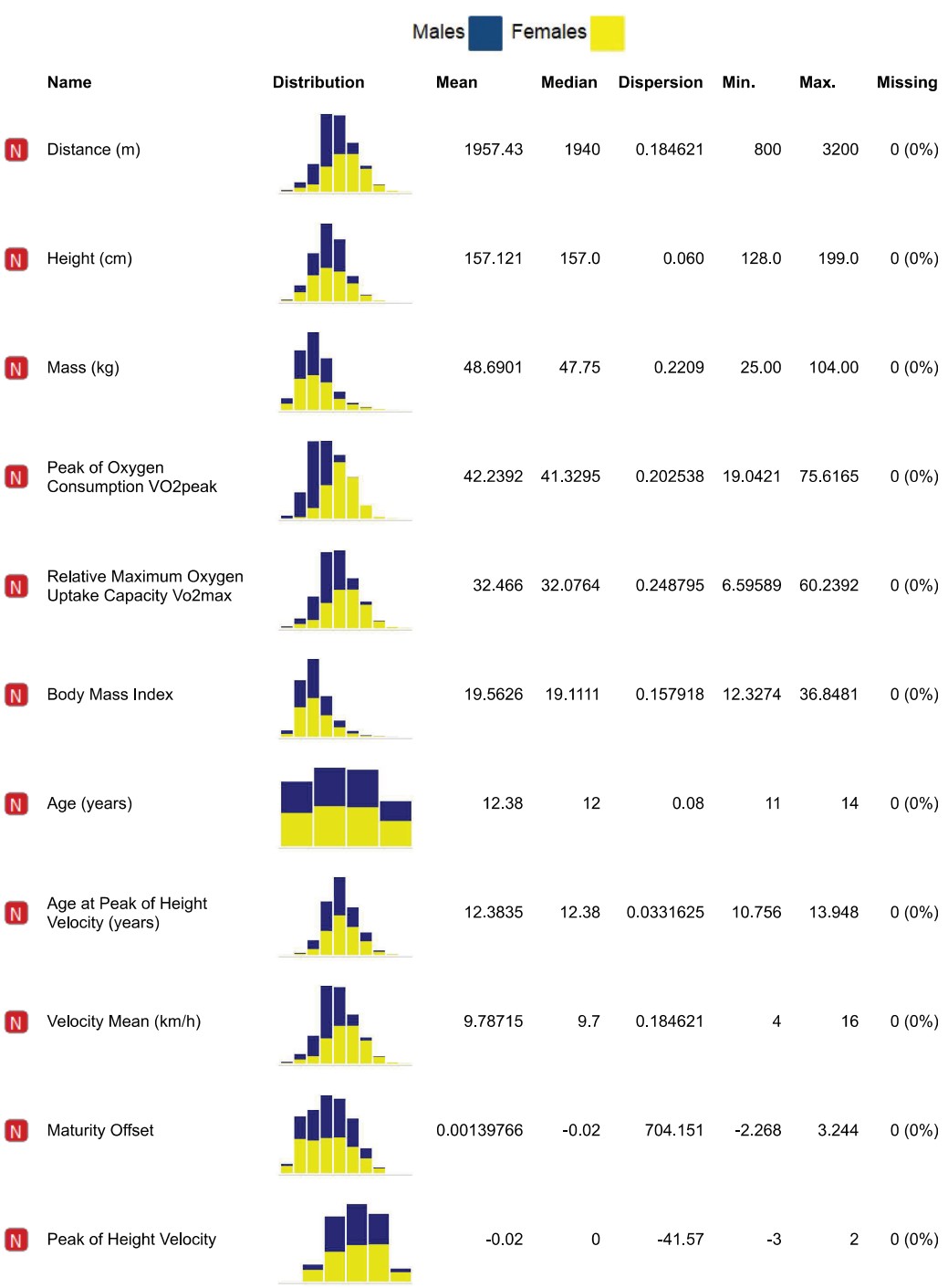

Figure 1 **Distribution of the parameters for girls (yellow), boys (blue) with related statistics.**

As represented in Fig. 2, CRT results for both sexes are fitted in the Gaussian Curve, it indicates that our sample was composed of athletic average-prepared young people, in fact we used strict inclusion criteria. We confirmed a mean better CRT result in boys *vs* girls, as observed in other studies (*Freitas et al., 2002*; *Lovecchio et al., 2013*); however, we
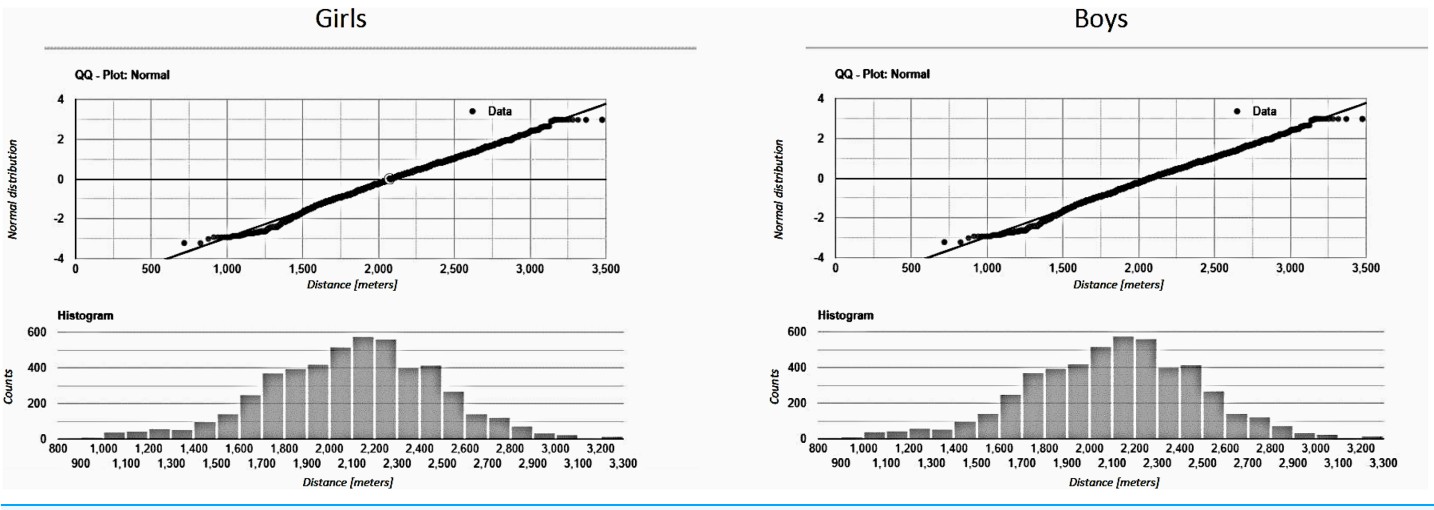

**Figure 2 Normality for males and females.**

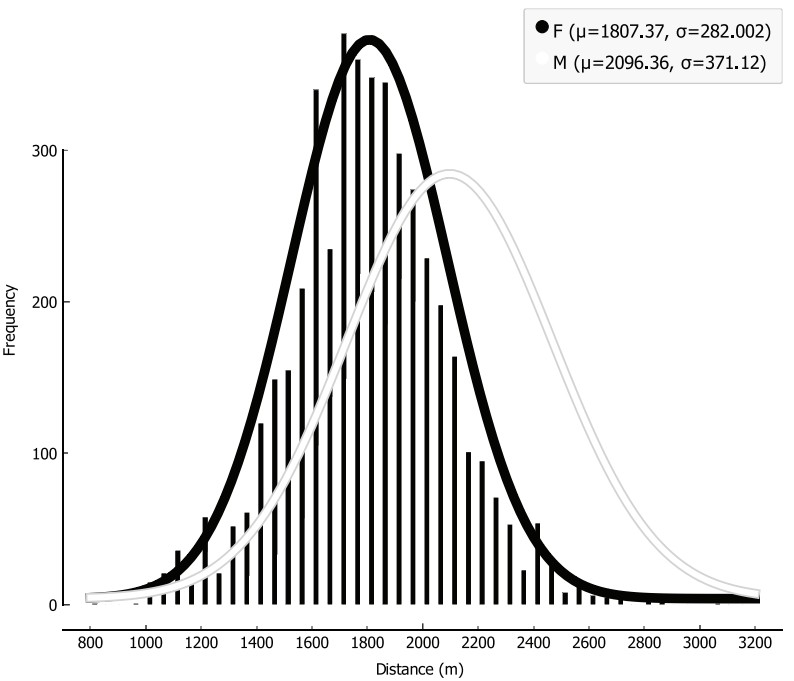

F (μ=1807.37, σ=282.002)
M (μ=2096.36, σ=371.12)

**Figure 3 Linear regression with body mass index (BMI) covariate.**

discovered a smaller SD in girls than boys, suggesting a more homogeneous aerobic status for girls that can be explained by the fact that they do less physical activity on average than boys in Italy, as described in the WHO Regional Office for Europe's report (*Inchley et al., 2020*). It is also very interesting that the Gaussian Distribution of CRT results suggests that aerobic capacity in adolescents (11–14 years old) is set in the middle, showing a normal distribution of physical activity status in Italian students of that age.

According to BMI, that is considered a simple resume of anthropometric characteristics (*Romero-Corral et al., 2008*) also in children (*Ceccarelli et al., 2020*), a homoscedastic

**Table 3 Ranking for univariate regression (dividing boys and girls) or R-Relief-F (including sex as covariate).**

| Regression parameters | Univariate regression males | Univariate regression females | R-Relief-F |
|---|---|---|---|
| BMI (Kg/m$^2$) | 245.023 | 509.784 | 0.024 |
| VO$_2$ peak (mL/kg/min) | 209.895 | 617.667 | 0.021 |
| Mass (Kg) | 132.925 | 164.218 | 0.021 |
| Age at PHV (years) | 4.116 | 150.476 | 0.018 |
| Height (cm) | 2.082 | 73.306 | 0.016 |
| PHV | 0.889 | 129.847 | 0.004 |
| Age (years) | 0.652 | 0.696 | 0.002 |
| Maturity offset | 0.001 | 150.476 | 0.011 |
| Sex | | | 0.000 |

**Note:**
BMI: body mass index; VO$_2$ peak: peak of oxygen consumption; Age at PHV, age at peak of height velocity; PHV, peak of height velocity.

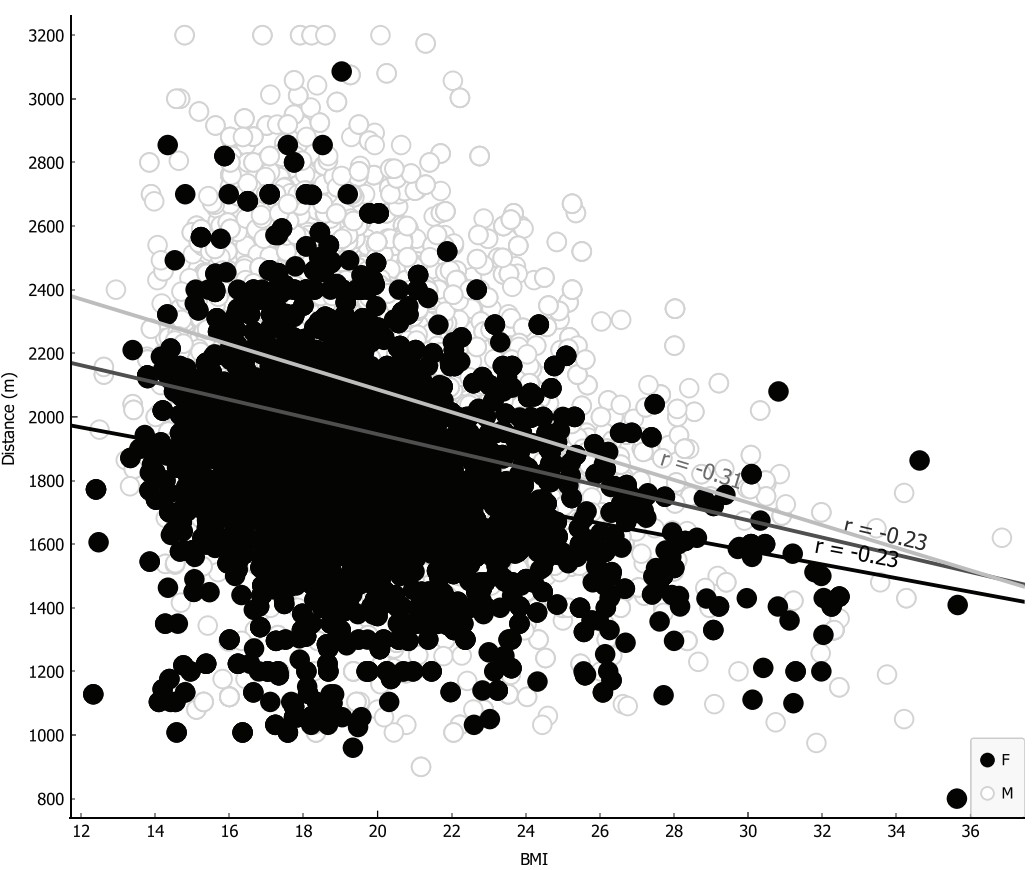

**Figure 4 Linear regression with body mass index covariate.**

distribution both for boys and girls is visually evident (Fig. 3); thus, showing difficult reasoning of inference according to height and mass, we can state the same result for the variable mass alone. In addition, analyzing linear correlation coefficients, we can affirm

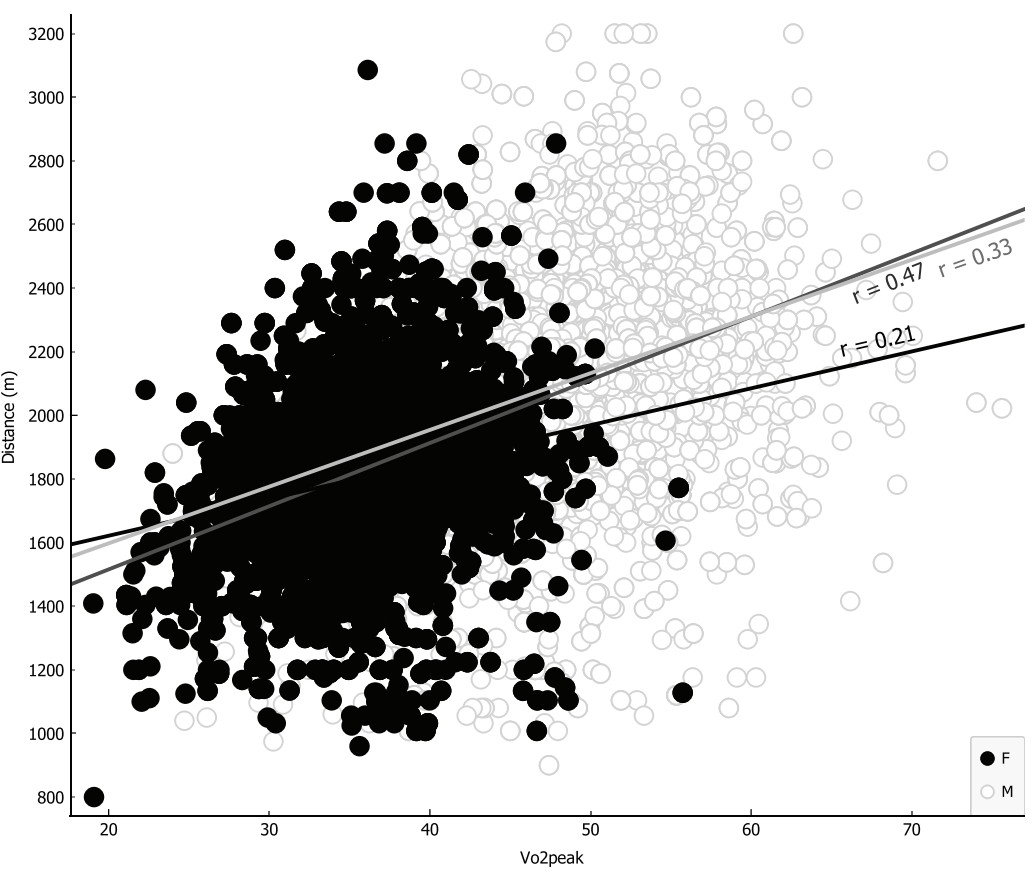

**Figure 5** **Linear regression with VO₂ peak covariate.**

there is a weak negative correlation for both BMI and mass variables with respect to the CRT results. We can explain this finding with the possible body composition differences when analyzing mass (and other related anthropometric parameters) in weight bearing performances, as suggested by other studies (*Artero et al., 2010*).

Figure 4 showed a visual homoscedastic distribution for VO₂ peak (calculated through an equation using height and mass), with a positive, but low linear correlation coefficient, leading to an accuracy reduction of indirect formulas to predict VO₂ peak in average adolescents. The only visual heteroscedastic distribution was observed in regression between distance in CRT and age at peak of height velocity (PHV), in addiction linear correlation coefficient was near 0 (0.01 for boys and −0.03 for girls) with a countertrend between sexes, according to the theory in which testosterone production determines improvements of performance (*Handelsman, Hirschberg & Bermon, 2018*), so we can affirm that height and mass are just indicators subjected to uncontrollable variations. Our study states further, for an unbiased population, the same that a recent paper finds that age at PHV is not a significant predictor in endurance performance prediction of basketball girls adolescents (*Gryko et al., 2022*).

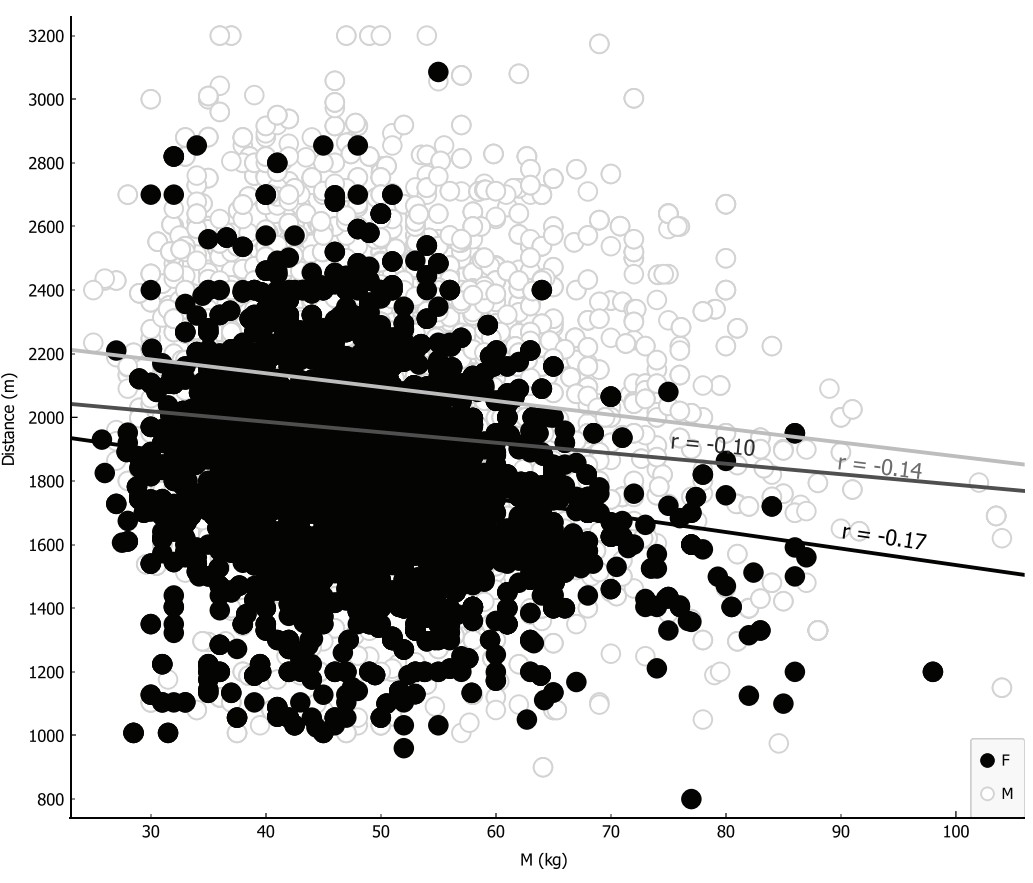

**Figure 6 Linear regression with mass covariate.**

Based on a non-linear analysis, *Armstrong & Welsman (2019a)* suggested that a sex-specific morphological covariate is more appropriate to investigate performance avoiding bias due to the maturity effects. Suggesting that the cross-sectional analyses in young adolescents show that there is no significant gender difference in maximal stroke volume once the fat-free mass has been correctly checked through multilevel modeling (*Armstrong & Welsman, 2019b*).

*Giuriato et al. (2020)* proposed a multiplicative allometric model to investigate CRF with CRT as dependent variable, showing that CRT performance during growth may not be predictable with only one anthropometry. Further, the results found by *Giuriato et al. (2020)* showed in both sexes an improvement in CRT distance. This trend using the scaling method, associated with anthropometrics (height, body mass), suggests that slim subjects correspond to the optimal height-to-body mass for CRF, highlighting that body mass (probably fat mass) influenced negatively the CRF performance, on the contrary height influenced the CRF performance positively. These results are in line with *Nevill et al. (1998)*, who suggested that proportional oxygen absorption peak and oxygen supply/use is facilitated by an increase in fat-free mass, thus promoting a lighter weight and penalizing heavier young people. Indeed, normalization only per body mass or height is conflicting, because growth does not follow a linear trend. In fact, *Armstrong & Welsman (2019a,*
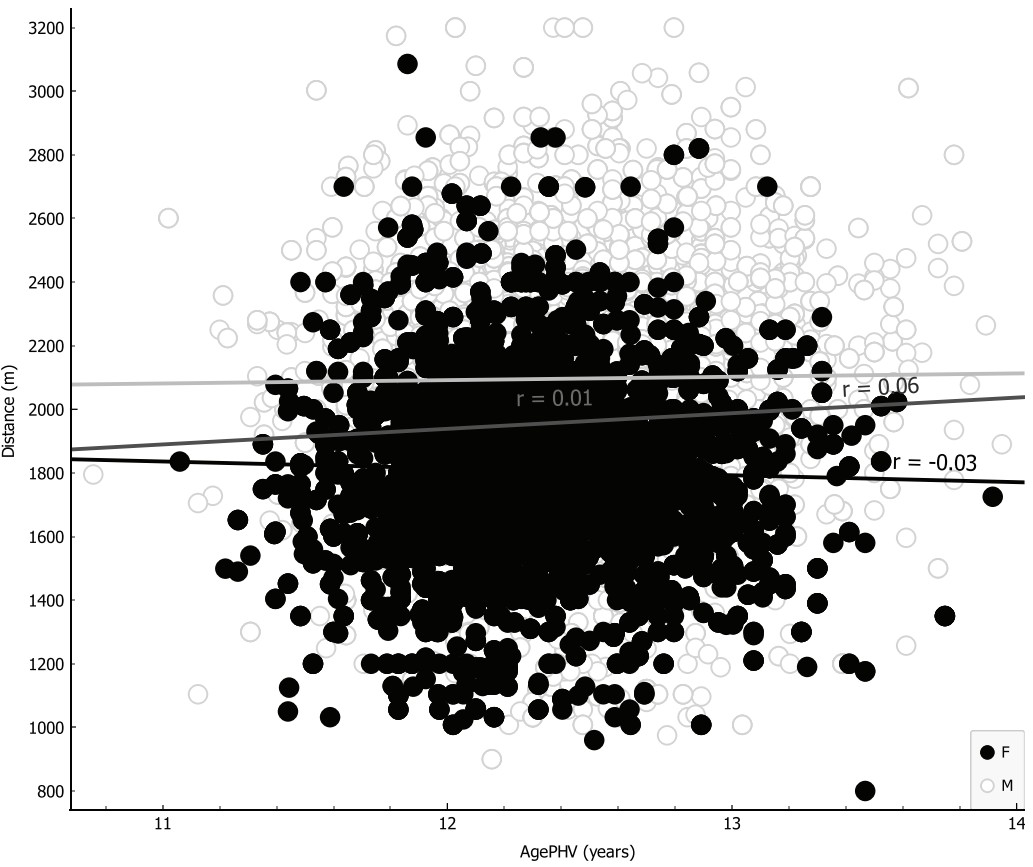

**Figure 7 Linear regression with age at peak of height velocity covariate.**

*2019b*) suggested that the VO$_2$ peak (deriving by the meter run) increases with simultaneous changes in morphological covariates specific to gender, age, and maturity, with the timing of these individual-specific changes. Further, *Lovecchio et al. (2023)*, using BMI-adjusted analyses, found a small to moderate decline in CRF between 1984 and 2010, proposing that these trends were probably affected by increases in sedentary behaviors and subsequently declines in vigorous physical activity levels.

Anthropometric variations during growth are non-linear and therefore confusing, indeed *Giuriato et al. (2020)* found a different correlation between body mass and CRF in adolescent boys and girls. Even Scandinavian data, collected by *Santtila et al. (2006)*, did not use weight gain as a reason for explaining the decline in CRF, despite the subjects being 19-year-old boys, and therefore out of the tumultuous period of growth. Thus, during growth body mass accentuates its instability making its use only unreliable.

## Limitations

We recognize that our study has some limitations. Firstly, because of the age range of our sample (11–14 years), our results could be influenced by the hormonal growth of these children, so future studies should consider a wider age range (*e.g.*, up to 18 years) to assess these trends while also considering full hormonal development. Moreover, it could be

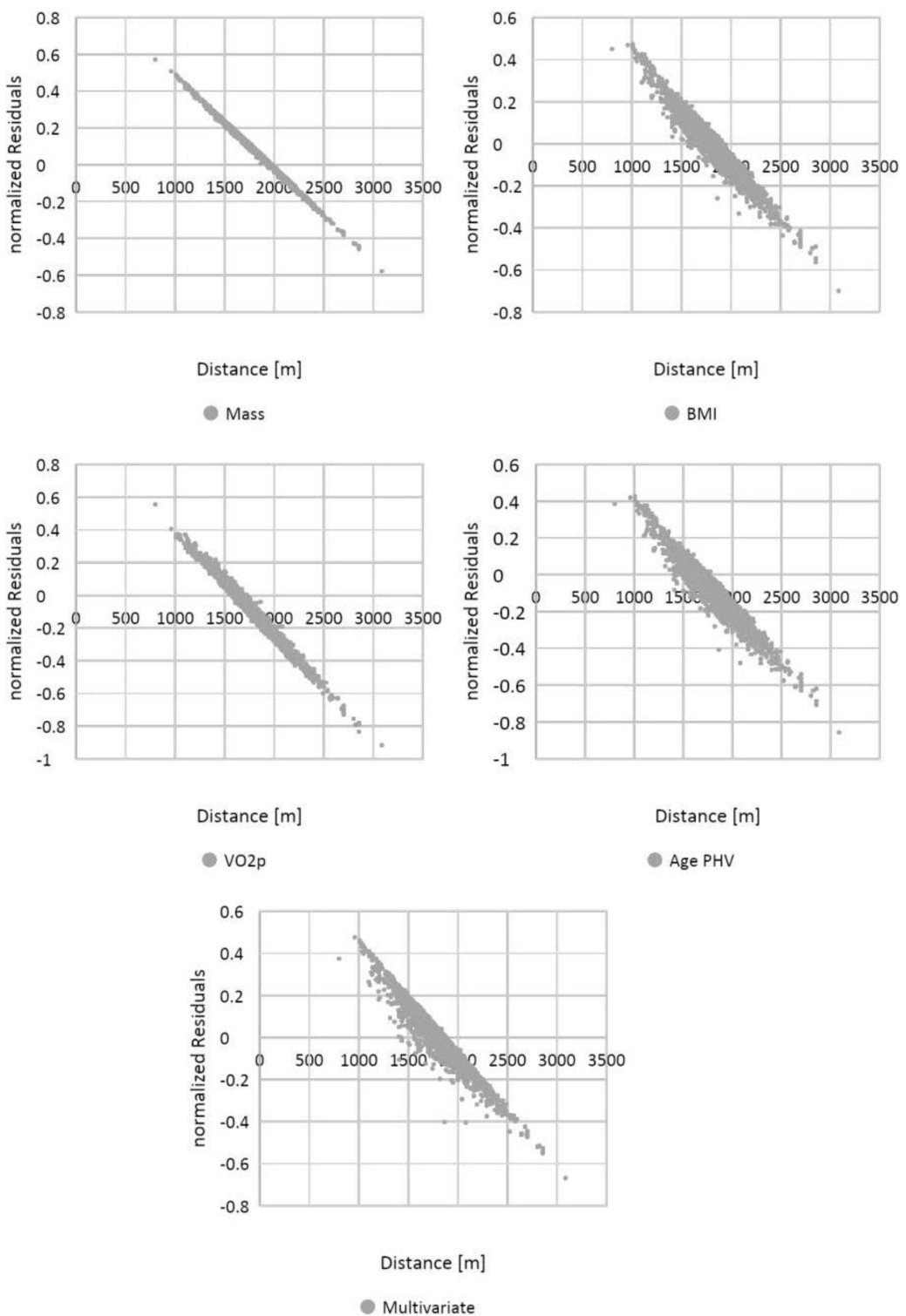

**Figure 8 Visual representation of homoskedasticity for covariates.**

useful to propose this test not in a designated track, but in a more realistic context in order to reach a larger number of children.

### Strengths

Our results showed that the adolescents' performance estimation based on anthropometric characteristics needs to be critically considered and that both anthropometric assessment and field tests need to be repeated over time to obtain more accurate results. In addition, it guides the approach needed for organizing training lessons by dividing groups of students for their proven athletic level and not for their anthropometric measures. In this light, of our study suggests that teachers can be confident in using repeated over time tests to assess students' performance regardless of the anthropometric outcomes such as height and weight that are affected by tumultuous growth phases.

## CONCLUSIONS

In conclusion, CRT results showed a normal distribution for boys and girls independently of their anthropometric measures and a very weak or absent correlation for all the parameters and leading to the assertion that CRT results cannot be predicted from anthropometric data.

### Funding

The authors received no funding for this work.

### Competing Interests

Vittoria Carnevale Pellino & Matteo Vandoni are Academic Editors for PeerJ.

### Author Contributions

- Gianluca Azzali conceived and designed the experiments, performed the experiments, prepared figures and/or tables, and approved the final draft.
- Massimo Bellato conceived and designed the experiments, performed the experiments, prepared figures and/or tables, and approved the final draft.
- Matteo Giuriato conceived and designed the experiments, prepared figures and/or tables, and approved the final draft.
- Vittoria Carnevale Pellino analyzed the data, prepared figures and/or tables, and approved the final draft.
- Matteo Vandoni analyzed the data, authored or reviewed drafts of the article, and approved the final draft.
- Gabriele Ceccarelli conceived and designed the experiments, analyzed the data, authored or reviewed drafts of the article, and approved the final draft.
- Nicola Lovecchio analyzed the data, authored or reviewed drafts of the article, and approved the final draft.

## Human Ethics

The following information was supplied relating to ethical approvals (*i.e.*, approving body and any reference numbers):

The institutional review board of Regione Lombardia along with the Italian National Olympic Committee (CONI).

## Data Availability

The raw measurements are available in the Supplemental File.

## Supplemental Information

Supplemental information for this article can be found online at http://dx.doi.org/10.7717/peerj.15271#supplemental-information.

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
