# Peer review of "Are anthropometric characteristics powerful markers to predict the Cooper Run Test? Actual Caucasian data"

_PeerJ, doi:10.7717/peerj.15271_

## Round 0.1 · original submission · Major Revisions

Thank you for submitting the manuscript to PeerJ. It has been reviewed by experts in the field and we request that you make major revisions before it is processed further.

We look forward to hearing from you soon.

Best wishes,

Badicu Georgian, Ph.D

Reviewer 1 ·

Basic reporting

The references are up-to-date, and the structure of the article is enough. Although, the replicability of the methods should be improved.

Experimental design

The description of the data collection context is missing. More information should be added:

Line 120: provide some context before. Namely introduce sampling strategy, power size estimation, and the selection process—additional information about the eligibility criteria and recruitment forms is also required.

Line 134: provide context about data collection. On which day of the week was performed? How many hours of rest were ensured? How was the recovery of the participants before? What time before the last meal and the test? Did the students have familiarization with the test? How? What is the sequence of tests? Did they perform a warm-up before Cooper’s test? Were the anthropometric assessments made before or after Cooper’s trial?

Validity of the findings

Line 148: provide statistics about the validity and reliability of the test for this type of population.

Line 179: did the authors have any chance to add the normality and homogeneity level directly?

Lines 183-184: did the assumptions for running the regression ensure?

Line 189: The text should present descriptive information about the main results coming from regression.

Additional comments

ABSTRACT

Line 50: replace "weight." Weight is the effect of the gravitational pull on the object's mass (measured in Newtons).

Line 51: provide some details about the experimental approach (namely, days of the assessment and sequence of measurements)

Lines 52-56: provide statistics (p; percentage of difference) that can sustain the sentences

Lines 59-61: The conclusions do not directly answer the study objective. It would be essential to reach a conclusion that directly answers the main research question.


INTRODUCTION

Line 86: it would be essential to add information about typical values of validity and reliability of the tests for the measurements performed in different age groups or competitive levels. Maybe it should appear before describing specific values in the next paragraph.

Line 105: at this stage, it would be necessary to a summary of evidence. The article-by-article description is interesting, but a summary of evidence from the articles mentioned above is needed.

Line 111: before the Italy context, it would be essential to describe the context of previous studies that have used anthropometry to standardize the outcome and the main equations extracted from that.

METHODS

Line 120: provide some context before. Namely introduce sampling strategy, power size estimation, and the selection process—additional information about the eligibility criteria and recruitment forms is also required.

Line 134: provide context about data collection. On which day of the week was performed? How many hours of rest were ensured? How was the recovery of the participants before? What time before the last meal and the test? Did the students have familiarization with the test? How? What is the sequence of tests? Did they perform a warm-up before Cooper's test? Were the anthropometric assessments made before or after Cooper's trial?

Line 139: provide statistics about the accuracy of the observers.

Line 148: provide statistics about the validity and reliability of the test for this type of population.

Line 156: indicate which is the primary outcome collected from the test.

Line 159: weight? Or body mass?

Line 174: the description of statistical analysis is missing. Must present the tests used, the values of normality and homogeneity, and the other assumptions needed to run some tests.

RESULTS

Line 179: did the authors have any chance to add the normality and homogeneity level directly?

Lines 183-184: did the assumptions for running the regression ensure?

Line 189: The text should present descriptive information about the main results coming from regression.


DISCUSSION

Line 230: study limitations are missing. Practical implications are missing. Future research is missing.

·

Basic reporting

The article is methodologically consistent. However, the material and methods and results need to be improved.

It has relevant literature to support the research. However, more current references could be added in the introduction to strengthen the background.

The tables and figures are suitable for the presentation of the results, while the results of this research allow to have current data in relation to the topic addressed.

Some general observations are detailed below.

a) Review the objectives of the study. I suggest using concepts similar to those indicated in the title and in the methodology. In fact, in methodology it is clear which are the main variables: anthropometric means and Endurance Cooper test.

Experimental design

The design is relevant to the study. But you should clarify the following aspects:

a) It is suggested to write the eligibility criteria more clearly. Specifically, the following paragraph: "Sedentary students were considered to avoid training related influences while young with known chronic cardiac, respiratory, neurological, or musculoskeletal disorders were nonetheless excluded." This text would improve if the inclusion and exclusion criteria were made more explicit. Mention if they used any questionnaire to determine that the children were sedentary.

b) Indicate the brand and model of the instruments used in the measurements of height and body weight.

c) It is suggested to describe the measurements in greater detail. Being few, a precise description of these is expected to allow them to be replicated by other researchers and clinical professionals.

d) What is described in data analysis must be included in the previous items of the methodology or modify the name of the subtitle. This is because it is expected that in data analysis there will be a description of the statistical analysis performed. In fact, this item should be named statistical analysis. Therefore, in addition to modifying what was described above, it is requested to add a statistical analysis item where each statistical analysis procedure is specifically indicated. This aspect is so deficient that the level of significance used in the research is not even indicated.

Validity of the findings

The results are potentially attractive and relevant. However, the wording of the results needs improvement.

a) A more scientific description of the results is requested, using the p value when necessary. (see other articles in the journal).

b) Review the figures. These contain results of R values (which could also be included in the description of the results) that in some figures are seen upside down.

Additional comments

There are some errors in citations throughout the text. Correct according to what is requested in the format of the journal. (example: (D. M. Cooper et al., 1984) / K. H. Cooper, 1968).

Reviewer 3 ·

Basic reporting

No comment.

Experimental design

No comment.

Validity of the findings

No comment.

Additional comments

General comments:
The authors performed a study in which they tried to explore if there is the possibility to predict Cooper Run Test results based on anthropometric measurements. The study embraced almost 10000 children from different schools. Although I have to compliment the authors for such an important sample size, there are several important downsides to this article. Firstly, the introduction is not well written, there is no good lead-in (I would not expect the aim that the authors formulated based on everything written in the introduction). Authors exhaustively reported distances achieved based on the previous studies for males and females, which is out of the scope and unnecessary. Cooper test is a widely famous test and authors should not focus on writing about reported test results but to explain where is the necessity of predicting Cooper Test results based on anthropometric data. Also, there is no clear evidence reported in the introduction regarding this possibility. Cooper test predicts cardiorespiratory fitness, but why do authors think (where is reasoning?) that supports proposing an aim that anthropometric characteristics could predict Cooper test results and thus cardiorespiratory fitness? Furthermore, the method section does not contain a statistical analysis section. Results are poorly described, while the discussion section contains parts with reporting results, and very little real discussion (i.e., comparing the results of the present study with the studies of similar design, etc.). Conclusions have nothing to do with the aim of this study.

Specific comments:
Lines 82-83: Please rephrase the following sentence: “Among this plethora of tests, Cooper.s 12-minute Run Test (CRT) is considered appropriate from childhood to maturity (Ainsworth et al., 2000; Ayán et al., 2015; Penry et al., 2011) and of wide use: in particular, for the easiest set/procedure required.” It is not clear what is set/procedure in this context, and please avoid “:” in the middle of the sentence.

Line 98: Bad formatting.

Pages 101 and 102: Authors did not abbreviate previously F and M, and therefore these abbreviations should not be used.
Lines 99-105: Between these two lines, the authors wrote two paragraphs, one that contains two sentences, and the other that contains only one. It is not advisable to write such short paragraphs and advised to have paragraphs of similar length throughout the manuscript.
Lines 105-117: Same here, three paragraphs.
Lines 125-126: Authors abbreviate Physical Education (PE) for the second time.
The data analysis section is crowded. Formulas used to calculate derived dependent variables should be placed in a table or differently organized for easier follow-up.
There is no Statistical analysis section in which authors should explain how they treated dependent variables.
Gaussian distribution of the results is not something that is unexpected for the Cooper test results and should not be reported as an interesting finding. What is written as a second purpose in the discussion is in reality the primary aim of this study (Lines 202-203). Please be more consistent when writing.
The discussion of this study resembles more results section more than the actual discussion. Discussion should consist of comparing the results of this study with the findings obtained in the studies with a similar design (please see lines 213-217 as a typical representative of such a mistake).
The conclusions of this study are not relevant. Authors should conclude that there is no possibility to predict Cooper test results based on the anthropometric data. Instead, they are giving some recommendations that are outside the context.
It is advisable to present the results as means and standard deviations, rather than only averages (Table 2).
r coefficients are written upside down in Figures 3 and 5. Avoid using abbreviations in the title of the tables and figures.

---

## Round 0.2 · Major Revisions

Some changes are needed to the article. Please address them.

Best regards,

Reviewer 1 ·

Basic reporting

The authors performed a significant improvement in the manuscript. The introduction was also improved as well as the discussion. Congratulations.

Experimental design

Now the methods are replicable, and the caution regarding reliability and validity is absent.

Validity of the findings

Now the methods are replicable, and the caution regarding reliability and validity is absent.

·

Basic reporting

The study is methodologically consistent. Its objective is clear and is resolved through the results presented.

They use relevant literature for the topic that supports the scientific foundation of the manuscript.

Figures and tables are adequate and legible.

Experimental design

a) The calculation of the sample size is not clear. They refer to the use of clinical trials to obtain the power size estimation, which causes confusion. If your study is cross-sectional and of a predictive nature, you should make a sample calculation consistent with your design.

It is suggested to review previous studies of the journal and its way of making the calculation of the sample size explicit.

https://peerj.com/articles/14092/
https://peerj.com/articles/6416/
https://peerj.com/articles/5157/

b) The data analysis section should be replaced by statistical analysis. Review previously mentioned examples.

Validity of the findings

The results are described in a simple way. They could contain a more scientific form of presentation. However, it is possible to exhibit the results that are expected according to the objective.

Reviewer 3 ·

Basic reporting

I stick to my comments from the first revision.

Experimental design

I stick to my comments from the first revision.

Validity of the findings

I stick to my comments from the first revision.

Additional comments

I still think that this article should be rejected since it has fundamental flaws.

---

## Round 0.3 · Major Revisions

Some major changes are needed:

Introduction: As was also requested by one of the reviewers, it is necessary that the authors explain a lot better where is the necessity of predicting Cooper Test results based on anthropometric data.

Additionally, highlighting by the authors, as clearly as possible, the novelty of this study and the presentation of the hypothesis/s, is imperatively necessary.

For Table 1- please enter below the table, the full name of the abbreviations used, i.e. BMI, VO2, also for Table 3, Figures 1, 2 (are not very clearly understood, the quality of the image is low, also for these figures insert the abbreviations).

Discussion - It is good to present at the beginning of this chapter what was the purpose of this research.

Also, the discussion chapter should consist of comparing the results of this research with the findings obtained in the studies with a similar design. Please expand this chapter with this information and present the limits of the work and the practical implications.

For the Conclusion chapter, the authors should conclude that there is no possibility to predict Cooper test results based on the anthropometric data.



Georgian Badicu
Academic Editor
PeerJ Life & Environment

---

## Round 0.4 · Major Revisions

Some major changes are needed:

Introduction: As was also requested by one of the reviewers, it is necessary that the authors explain a lot better where is the necessity of predicting Cooper Test results based on anthropometric data.

These additions must be highlighted in the paper!!!!!!

Additionally, highlighting by the authors, as clearly as possible, the novelty of this study and the presentation of the hypothesis/s, is imperatively necessary.

These additions must be highlighted in the paper!!!!!!


For Table 1- please enter below the table, the full name of the abbreviations used, i.e. BMI, VO2.

These additions must be added below the tables!!!!

Discussion - It is good to present at the beginning of this chapter what was the purpose of this research.

These additions must be highlighted in the paper!!!!!!

Also, the discussion chapter should consist of comparing the results of this research with the findings obtained in the studies with a similar design. Please expand this chapter with this information and present the limits of the work and the practical implications.

These additions must be highlighted in the paper!!!!!!

For the Conclusion chapter, the authors should conclude that there is no possibility to predict Cooper test results based on the anthropometric data.

These additions must be highlighted in the paper!!!!!!

Georgian Badicu
Academic Editor
PeerJ Life & Environment

---

## Round 0.5 · Minor Revisions

Please refer to the text where is the answer, preferably to cross the lines, e.g see line/s.....

Statistical analysis - which statistical program did you use and which version? Please fill in the necessary information.

For Table 1- please enter below the table, the full name of the abbreviations used, i.e. BMI, VO2, also for Table 3, not inside the table, below the table.

The quality of Figure number 2 is not improved enough, please redo it.

Conclusions: ....''confirming the authors' hypothesis, and leading to the assertion that CRT results cannot be predicted from anthropometric data.'', what hypothesis are we referring to? I did not see any hypothesis in the paper, can you be more clear about this statement?

---

## Round 0.6 · Minor Revisions

The paper is almost ready to be accepted for publication. However, the Section Editor noted that there are further minor edits and comments the authors need to address before finally recommending the manuscript to be published. Please see the attached annotated file.

With kind regards,

Georgian Badicu
Academic Editor
PeerJ Life & Environment

---

## Round 0.7 · Minor Revisions

Please address the Section Editor's revisions in the attached PDF.

Thank you!

With kind regards,
Georgian Badicu
Academic Editor
PeerJ Life & Environment

---

## Round 0.8 · accepted · Accept

Dr Azzali and colleagues,

The original Academic Editor is no longer available so I have taken over handling the submission in my capacity as Section Editor.

Thank you for addressing all of the Reviewers' edits and comments in a timely manner. I am pleased to recommend your amended manuscript for publication.

Thank you for supporting PeerJ and we look forward to receiving future manuscripts from you and your research team. Thanks, A/Prof Mike Climstein